# Genome-Wide Identification and Expression Analysis Reveals Roles of the *NRAMP* Gene Family in Iron/Cadmium Interactions in Peanut

**DOI:** 10.3390/ijms24021713

**Published:** 2023-01-15

**Authors:** Zengjing Tan, Jinxiu Li, Junhua Guan, Chaohui Wang, Zheng Zhang, Gangrong Shi

**Affiliations:** College of Life Sciences, Huaibei Normal University, Huaibei 235000, China

**Keywords:** *Arachis hypogaea*, NRAMP, iron deficiency, cadmium uptake and transport, cultivar difference

## Abstract

The natural resistance-associated macrophage protein (NRAMP) family plays crucial roles in metal uptake and transport in plants. However, little is known about their functions in peanut. To understand the roles of *AhNRAMP* genes in iron/cadmium interactions in peanut, genome-wide identification and bioinformatics analysis was performed. A total of 15 *AhNRAMP* genes were identified from the peanut genome, including seven gene pairs derived from whole-genome duplication and a segmental duplicated gene. AhNRAMP proteins were divided into two distinct subfamilies. Subfamily I contains eight acid proteins with a specific conserved motif 7, which were predicted to localize in the vacuole membrane, while subfamily II includes seven basic proteins sharing specific conserved motif 10, which were localized to the plasma membrane. Subfamily I genes contained four exons, while subfamily II had 13 exons. AhNRAMP proteins are perfectly modeled on the 5m94.1.A template, suggesting a role in metal transport. Most *AhNRAMP* genes are preferentially expressed in roots, stamens, or developing seeds. In roots, the expression of most *AhNRAMP*s is induced by iron deficiency and positively correlated with cadmium accumulation, indicating crucial roles in iron/cadmium interactions. The findings provide essential information to understand the functions of *AhNRAMP*s in the iron/cadmium interactions in peanuts.

## 1. Introduction

Iron (Fe) is one of the essential micronutrients for virtually all organisms. Due to its ability to alternate between ferric (Fe^3+^) and ferrous (Fe^2+^) ions, Fe acts as a structural cofactor in many enzymes and plays important roles in many metabolic processes such as photosynthesis, respiration, sulfur assimilation, and nitrogen fixation [1]. The shortage of Fe can inhibit chlorophyll synthesis, reduce photosynthesis, and interrupt the respiratory electron transport and tricarboxylic acid cycle [2]. Although Fe is abundant in soils, its bioavailability is limited because it mainly exists in insoluble ferric hydroxides [3]. Fe deficiency in the edible tissues of plants is a critical issue for human health as most people rely on plant-based diets for their primary Fe source. Therefore, understanding the physiological and molecular mechanisms of Fe uptake and translocation in plants is critical to improving their capacity for Fe acquirement.

Cadmium (Cd) is a non-essential trace metal with high toxicity to nearly all living organisms. Since there are no specific transporters for Cd in plant cells, it enters into roots and transfers to the aerial parts by hijacking the transport pathways of micronutrients, including Fe [4]. Therefore, there is an interaction between Cd and Fe in the process of uptake and transport. Cd exposure significantly reduces Fe concentrations in plants [5]. Fe deficiency increased the uptake and accumulation of Cd [6,7,8], while excess Fe reduced Cd uptake in plants [5,9].

The NRAMPs (natural resistance-associated macrophage proteins) are the integral membrane transporters that play an essential role in divalent metal transport across the cellular membrane. NRAMP proteins are highly conserved from bacteria to mammals, with 10–12 transmembrane domains (TMD) and a consensus transport signature [10]. In *Arabidopsis*, six members (NRAMP1-6) were identified, which are classified into two subfamilies based on phylogenetic analysis [11]. AtNRAMP1 is a plasma membrane-localized transporter that acts as a high-affinity manganese (Mn) transporter in roots under low Mn conditions [12] and plays a pivotal role in Fe transport by cooperating with IRT1 [13]. AtNRAMP2 is a resident protein of the trans-Golgi network and likely exports Mn from the trans-Golgi network lumen to the cytosol under Mn-deficient conditions [14,15]. Both AtNRAMP3 and AtNRAMP4 are vacuolar metal transporter and redundantly mediate the vacuolar export of Fe and Mn under limiting conditions [16,17,18]. AtNRAMP6 functions as an intracellular metal transporter and are involved in cellular Cd distribution, contributing to Cd toxicity [19]. To date, functions of AtNRAMP5 have not been characterized in detail.

Rice (*Oryza sativa*) has seven *NRAMP* members, and most of them have been functionally characterized. OsNRAMP5 is a plasma membrane-localized transporter that contributes to the uptake of Fe, Mn, Cd, and lead (Pb) [20,21,22,23,24,25]. Knockout of *OsNRAMP5* significantly reduces concentrations of Mn and Cd in the roots, shoots, and grains [22,24] but simultaneously facilitates Cd translocation from roots to shoots [21]. Overexpression of *OsNRAMP5* markedly decreased root-to-shoot Cd translocation by disrupting its radial transport into the stele for xylem loading [20]. OsNRAMP1, a close homolog of OsNRAMP5 (72.8% similarity in the amino acid sequences), is localized to the plasma membrane of root and leaf cells, having similar functions with OsNRAMP5 but not redundant [26,27,28]. OsNRAMP2 is a tonoplast-localized transporter that transports Fe from the vacuole to the cytosol [29]. OsNRAMP3 is a vascular bundles-specific Mn transporter that is highly expressed in the node and preferentially transports Mn to young leaves under Mn-limiting conditions [30,31]. OsNRAMP4 (also known as Nrat1) is a plasma membrane-localized transporter for aluminum (Al) but not for divalent cations in yeast [32,33]. OsNRAMP6 is a plasma membrane-localized transporter that functions as iron and Mn transporters and contributes to disease resistance [34].

Peanut (*Arachis hypogaea* L.) is an allotetraploid (2n = 4x = 40) legume widely cultivated in temperate and tropical regions of the world for edible oil production and digestible protein sources. However, peanut is sensitive to iron deficiency and frequently suffers from Fe deficiency chlorosis in calcareous soil, which greatly limits the yield and quality [35]. More seriously, peanut has a high ability to accumulate Cd in seeds and vegetative tissues [36,37], and iron deficiency significantly increases the absorption and accumulation of Cd in peanut plants [6,7,38]. Fe deficiency chlorosis and Cd pollution have become important constraints in peanut cultivation. In order to avoid these adverse effects, it is necessary to understand the mechanism of Fe/Cd interaction during their uptake and translocation.

Although *AhNRAMP1* has been confirmed to be involved in Fe accumulation in young leaves and tolerance to Fe deprivation [39], little data is available regarding the *NRAMP* gene family in peanut. Here, 15 *NRAMP* genes were identified from the peanut genome, and their structure, function, and evolution were analyzed. Additionally, the expression of *NRAMP* genes in response to Fe deficiency and Cd exposure was evaluated. These findings would provide clues to further characterize the functions of *AhNRAMP* genes and to understand the mechanisms of Fe/Cd interaction in peanut plants.

## 2. Results

### 2.1. Identification and Phylogenetic Analysis of the AhNRAMP Family in Peanut

A total of 19 candidate sequences were identified by a BLASTP search against the peanut genome. Although all of them contain the NRAMP domain, four sequences were excluded because they are homologous with ethylene-insensitive proteins (EINs, Appendix A). Among the remaining 15 sequences, a short sequence (NCBI gene ID: 112717764) only contains the last five residues (MQGFL) of the typically conserved amino acid residues GQSSTITGTYAGQY(/F)V(/I)MQGFL [40]. However, we still retained it (named AhNRAMP1.2) because synteny analysis revealed that it is the segmental duplicated gene of AhNRAMP1.1. Thus, a total of 15 putative *AhNRAMP* genes were identified in peanut, and were named based on the phylogenetic relationships with *AtNRAMP*s from *Arabidopsis* (Table 1).

To elucidate the phylogenetic distribution of AhNRAMPs, a tree was constructed based on 58 NRAMP protein sequences from peanut, *A. duranensis*, *A. ipaensis*, *Medicago truncatula*, soybean, common bean, *Arabidopsis*, and rice. The tree topology showed that the 58 NRAMP members were divided into two distinct subfamilies: subfamily I and II (Figure 1). Each subfamily contained members from different species, suggesting close genetic conservation among NRAMPs. Subfamily I, typified by the AtNRAMP2/3/4/5, contained eight AhNRAMP members. Subfamily II is typified by the AtNRAMP1/6 and consists of seven AhNRAMP members (Figure 1). Notably, NRAMP members from the six legumes are located on adjacent branches with good genetic relationships.

The length of *AhNRAMP* genes varied from 742 bp (*AhNRAMP1.2*) to 2505 bp (*AhNRAMP6.3*), with CDS lengths from 570 bp (*AhNRAMP1.2*) to 1680 bp (*AhNRAMP2.2*/*2.4*). The amino acid number of AhNRAMP proteins ranged from 190 (AhNRAMP1.2) to 560 (AhNRAMP2.2/2.4), and the molecular weight varied from 20.83 kDa (AhNRAMP1.2) to 61.39 kDa (AhNRAMP2.4). The instability, aliphatic index, and GRAVY of AhNRAMP proteins ranged from 31.17 (AhNRAMP1.2) to 42.40 (AhNRAMP2.3), from 108.53 (AhNRAMP2.2) to 124.19 (AhNRAMP3.4), and from 0.355 (AhNRAMP2.2) to 0.693 (AhNRAMP3.4), respectively. The isoelectric point (pI) of subfamily I members less than 7, while that of subfamily II members is larger than 7 (Table 1). All AhNRAMP proteins contained 12 TMDs except AhNRAMP1.2, which has only four TMDs (Table 1). The eight subfamily I proteins were predicted to localize in the vacuole membrane, whereas the seven subfamily II proteins localized to the plasma membrane (Table 1).

### 2.2. Conserved Motifs, Domains, and Models of AhNRAMP Proteins

A total of ten conserved motifs were identified in the sequences of AhNRAMP proteins, and all of them were annotated to be *NRAMP* according to the Pfam tool (Figure 2A and Appendix A). As shown in Figure 2A, all AhNRAMP proteins contain nine motifs except AhNRAMP1.2. The members of the same subfamily shared the same motif composition; however, considerable differences were also observed between the two subfamilies. Subfamily I members specifically contained motif 7, while subfamily II specifically contained motif 10. All AhNRAMP proteins shared motifs 1, 2, 3, 4, 5, 6, 8, and 9 except AhNRAMP1.2, which only contains the last two motifs of AhNRAMPs, indicating a distinct evolutionary process and physiological function. All AhNRAMP proteins contained one domain named NRAMP, which is the typical domain of the NRAMP family (Figure 2B).

### 2.3. Exon-Intron Structure, Duplication, and Ka/Ks of the AhNRAMP Family

The two subfamilies differed from each other in the exon-intron structure (Figure 2C). *AhNRAMP* genes belonging to subfamily I contained four exons with three introns, while those of subfamily II had 13 exons except *AhNRAMP1.2*, which had three exons and three introns (Figure 2C). However, *AhNRAMP* genes that clustered in the same branch of the evolutionary tree are similar in the exon-intron organization.

Fifteen *AhNRAMP* genes are distributed unevenly on 13 chromosomes of the two subgenomes, and the subgenome A (Chr.01–10) and B (Chr.11–20) contained eight and seven *AhNRAMP* genes, respectively (Figure 3A and Appendix A). Chromosomes 08 and 17 contained two *AhNRAMP* genes, respectively, while no *AhNRAMP* gene was distributed on chromosomes 03, 04, 06, 13, 14, 16, and 20, and each of the remaining 11 chromosomes contained only one gene. All *AhNRAMP* genes experienced duplication events, including seven pairs of whole-genome duplication (WGD) genes (*AhNRAMP1.1*/*1.3*, *AhNRAMP2.1*/*2.3*, *AhNRAMP2.2*/*2.4*, *AhNRAMP3.1*/*3.3*, *AhNRAMP3.2*/*3.4*, *AhNRAMP6.1*/*6.3*, and *AhNRAMP6.2*/*6.4*) and five pairs of the segmental duplicated genes (*AhNRAMP1.1*/*1.2*, *AhNRAMP3.1*/*3.2*, *AhNRAMP3.3*/*3.4*, *AhNRAMP6.1*/*6.2*, and *AhNRAMP6.3*/*6.4*) (Figure 3A). No tandem duplication was identified in *AhNRAMP* genes.

To better understand the evolutionary process of the *AhNRAMP* family, a collinear map was constructed between the two ancestral species, *A. duranensis* and *A. ipaensis* (Figure 3B and Appendix A). As presented in Figure 3B, there were five orthologous gene pairs between *A. duranensis* and *A. ipaensis*, which are less than that of WGD-derived genes in peanut (seven gene pairs). Compared with the peanut genome, a homologous gene of *AdNRAMP2* was lost on chromosome A08 of the *A. duranensis* genome, and a homologous gene of *AiNRAMP1* was lost on chromosome B01 of the *A. ipaensis* genome. Moreover, *AiNRAMP6.1*, which was originally located on chromosome B08, was translocated to chromosome B09 in the *A. ipaensis* genome (Figure 3B).

To further reveal the evolution of the *NRAMP* family, a comparative syntenic analysis was carried out on peanut, *A. duranensis*, *A. ipaensis*, common beans, and *Arabidopsis*. A total of 23 and 18 collinear blocks were detected between peanut (15 genes) and *A. duranensis* (7 genes) and between peanut (14 genes) and *A. ipaensis* (6 genes), respectively (Figure 3C). All *AhNRAMP* genes were collinear with the seven *PvNRAMPs* of the common bean through the seven *AdNRAMP*, suggesting a close evolutionary relationship among the three species. Four *PvNRAMPs* of common bean showed a collinear relationship with five *AtNRAMP* genes of *Arabidopsis*, while the other three were not collinear with *AtNRAMP* genes (Figure 3C).

The *Ka*/*Ks* ratios of all duplicated pairs (0.01–0.90) were less than 1 (Table 2), indicating a purifying selection in the evolutionary process of *AhNRAMP* genes [41]. The divergence time for the ten whole-genome duplicated gene pairs ranged from 0.000 Mya to 3.697 Mya, which was dramatically less than most of the segmental duplicated genes (51.236–58.221 Mya) (Table 2).

### 2.4. 3D Model Predictions and Multiple Sequence Alignment of AhNRAMP Proteins

All AhNRAMP proteins were well modeled with the template, 5m94.1.A (Figure 4 and Appendix A). Sequence identity ranged from 36.52% to 42.19%, the value of GMQE ranged from 0.32 to 0.59, and the QMEANDisCo global score ranged from 0.63 to 0.70 (Appendix A). These data suggest a high quality of 3D model predictions for AhNRAMP proteins.

Multiple sequence alignment showed considerable homology between the ScaDMT and AhNRAMP proteins throughout the TMDs (Figure 5). Compared with ScaDMT, which has 11 TMDs, an additional TMD (TM12) was harbored in the C terminus of AhNRAMP proteins. All AhNRAMPs contained typical conserved amino acid residues GQSSTITGTYAGQY(/F)V(/I)MQ(/G/E)GFL except AhNRAMP1.2, in which the last five residues (MQGFL) were reserved. Besides, conserved amino acid residues were also found in different TMDs, e.g., DPGN in TM1, EVIGS(/T)AI(/L)A in TM3, M(/T)PHNV(/L) FLHSS(/A)LVQ(/L)SF in TM6, A(/G)LLAS(/A) in TM8 (Figure 5).

### 2.5. cis-Acting Elements of AhNRAMP Genes

To better understand the potential regulation of *AhNRAMP* genes, the *cis*-acting elements were predicted and shown in Appendix A. The promoter regions of all *AhNRAMP* genes harbored CAAT-box, TATA-box, and Box 4 (Appendix A). The number ranged from 23 to 43 for CAAT-box, from 45 to 137 for TATA-box, and from 1 to 12 for Box 4, respectively. Besides, a number of *cis*-acting elements such as ABRE, G-box, ARE, TCT-motif, TC-rich repeats, CGTCA-motif, TGACG-motif, TCA-element, AT1-motif, GATA-motif, GT1-motif, circadian, MBS, and LTR, were frequently identified in promoter regions of *AhNRAMP* genes. Most of them are involved in light, abiotic stress, and hormone responsiveness (Appendix A). 

### 2.6. Tissue-Specific Expression Profiles of AhNRAMP Genes

RNA-seq data showed that all *AhNRAMP* genes exhibited a tissue-specific expression in peanut plants, and their expression profiles were independent of phylogeny (Appendix A and Figure 6). Fifteen *AhNRAMP* genes could be classified into two distinct groups. Group 1 included *AhNRAMP1.2*, *AhNRAMP2.1*/*2.2*, *AhNRAMP3.2*/*3.3*/*3.4*, and *AhNRAMP6.4*, which show a low expression in most of the peanut tissues. Among them, *AhNRAMP3.3* was not or lowly expressed in all peanut tissues, *AhNRAMP3.2* and *AhNRAMP6.4* were mainly expressed in roots and seeds (Pattee 8/10), while the other four genes were preferentially expressed in the stamens. Group 2 is composed of the remaining eight genes representing high gene expression, and all of them were preferentially expressed in the roots and stamens. Besides, *AhNRAMP1.1* and *AhNRAMP2.4* also showed high expression in nodules, shoot tips, mainstem leaves, perianths, and developing seeds.

### 2.7. Gene Expression of AhNRAMPs in Response to Fe-Deficiency and Cd Exposure

To elucidate the responses of *AhNRAMP* genes to Fe-deficiency and Cd exposure, two contrasting peanut cultivars (Fenghua 1 and Silihong) were used for qRT-PCR analysis. As presented in Appendix A, compared with Silihong, the reduction of shoot dry weight and leaf chlorophyll contents (SPAD value) caused by Fe deficiency was more pronounced in Fenghua 1. Under Fe deficient conditions, Cd exposure resulted in larger decreases in shoot dry weight and leaf chlorophyll contents in Silihong compared with Fenghua 1 (Appendix A). The results, in agreement with the previous study [37], suggested that Silihong is more tolerant to Fe deficiency but more sensitive to Cd stress than Fenghua 1.

The expression of *AhNRAMP* genes differed between Silihong and Fenghua 1, and was significantly influenced by Fe-deficiency and/or Cd exposure (Figure 7). Except for *AhNRAMP2.4*, the expression of all genes in Silihong was higher than that of Fenghua 1 under normal nutrition. Fe deficiency up-regulated the expression of *AhNRAMP1.1*/*1.2*/*1.3*, *AhNRAMP2.4*, *AhNRAMP3.1*/*3.3*/*3.4*, and *AhNRAMP6.2* for both cultivars (Figure 7).

Under normal Fe supply, Cd exposure repressed the expression of *AhNRAMP2.2*, *AhNRAMP3.4,* and *AhNRAMP6.2*/*6.3*/*6.4* in Fenghua 1 but increased the expression of *AhNRAMP2.4* in Silihong. Under Fe-deficiency, Cd exposure repressed the expression of *AhNRAMP1.1*/*1.2*/*1.3*, *AhNRAMP2.4*, and *AhNRAMP6.2* but induced the expression of *AhNRAMP6.3* for both cultivars (Figure 7). The remaining genes showed a cultivar-specific response to Cd exposure under Fe-deficient conditions. Specifically, Cd-induced the expression of *AhNRAMP2.2*, *AhNRAMP3.2*/*3.3* in Silihong but reduced the expression of *AhNRAMP3.1*/*3.4* and *AhNRAMP6.2*/*6.4* in Fenghua 1 (Figure 7).

### 2.8. Fe/Cd Accumulation and Translocation in Two Peanut Cultivars

The two peanut cultivars showed a differential response in Fe accumulation to Fe deficiency (Figure 8). In Silihong, Fe deficiency significantly reduced Fe concentrations in roots and shoots, the total amount of Fe in plants, and the percentage of Fe in shoots. In Fenghua 1, Fe deficiency significantly reduced Fe concentrations in shoots and the percentage of Fe in shoots, while root Fe concentrations and the total amount of Fe in plants were not affected. Cd exposure increased Fe accumulation in peanut plants, which was more pronounced in the Fe-deficient treatment (Figure 8). In contrast, the percentage of Fe in shoots of Fenghua 1 was decreased by Cd exposure under normal Fe supply conditions.

Fe deficiency significantly enhanced root Cd concentrations and the total amount of Cd in plants for both cultivars, while the percentage of Cd in shoots was reduced (Figure 8). Shoot Cd concentrations in Silihong were also increased by Fe deficiency. Cd concentrations in roots were higher in Silihong than in Fenghua 1, regardless of Fe supply. Compared with Fenghua 1, higher shoot Cd concentrations and total amount of Cd in plants was also observed in Silihong under Fe-deficient conditions (Figure 8).

### 2.9. Relationship of Gene Expression of AhNRAMPs and Fe/Cd Accumulation

To identify the *AhNRAMP* genes involved in Fe/Cd accumulation, Pearson’s correlation analysis was performed. As shown in Table 3, the expression of *AhNRAMP1.1*/*1.2*/*1.3*, *AhNRAMP3.1*/*3.3*/*3.4*, and *AhNRAMP6.2* was significantly and negatively correlated with Fe concentrations in roots and shoots, as well as the total amount of Fe in plants. The expression of *AhNRAMP2.2*/*2.4* and *AhNRAMP6.3* was significantly correlated with the percentage of Fe in shoots. The expression of all *AhNRAMP* genes except *AhNRAMP2.4* and/or *AhNRAMP3.2* were significantly and positively correlated with Cd concentrations in roots and shoots, as well as the total amount of Cd in plants (Table 3). Negative correlations were observed between the expression of *AhNRAMP3.1*/*3.3* and the percentage of Cd in shoots (Table 3).

## 3. Discussion

Due to the important role in the uptake and transport of metal ions, a large number of *NRAMP* genes have been identified and functionally characterized [12,16,17,18,19,20,21,22,23,24,26,27,28,29,30,31,32,33,34]. However, there is a lack of detailed information about this family in peanuts. In this study, 15 *NRAMP* genes were identified from the peanut genome, which is the largest among reported plant species, including *Arabidopsis* (6) [11], rice (7) [42], soybean (13) [43], common bean (7) [44], potato (*Solanum tuberosum*, 5) [45], tea plant (*Camellia sinensis*, 11) [46], cacao (*Theobroma cacao*, 5) [47], and *Spirodela polyrhiza* (3) [48]. A large number of genes in the peanut genome have been reported in the gene families of metal tolerance proteins [49], zinc/iron-regulated transporter-like proteins [50], and oligopeptide transporters [51].

The expansion and functional diversification of gene families depend on gene duplication events, including WGD, segmental, and tandem duplication [48]. As an allotetraploid species, peanut contains two sets of subgenomes (A and B) from two ancestral species (*A. duranensis* and *A. ipaensis*) [52]. Syntenic analysis revealed that among the 15 *AhNRAMP* genes, seven paralogous gene pairs resulted from WGDs. WGD events are ubiquitous among angiosperms [53]. Peanut has experienced at least three rounds of such events together with the allopolyploidization [54]. Besides, three paralogous gene pairs in subgenome A (Chr. 01–10) and two paralogous gene pairs in subgenome B (Chr. 11–20) were found to be segmental duplicates. Similarly, the comparative syntenic analysis showed three and two pairs of segmental duplicated genes in the genome of *A. duranensis* (AA) and *A. ipaensis* (BB), respectively. These data suggest that segmental duplications of *AhNRAMP* genes have occurred before the allopolyploidization, which is illustrated by the considerably long divergence time of segmental duplicated genes (51.236–58.221 Mya). No tandem duplication was detected in peanut, *A. duranensis,* and *A. ipaensis*. Therefore, it might be WGD during allopolyploid speciation that contributes to the large number of *AhNRAMP* genes in peanut. 

*AhNRAMP* genes are unevenly distributed in the two subgenomes of peanut. Subgenome A contained 8 *AhNRAMP* genes, while subgenome B contained seven genes (Figure 3A). Meanwhile, the number of *AhNRAMP* genes differed between the two diploid progenitors, *A. duranensis* (7) and *A. ipaensis* (6) (Figure 3B). These differences suggest an asymmetrical evolution between the two subgenomes of peanut. Notably, the number of *AhNRAMP* genes in the peanut genomes was higher than the total number of genes in *A. duranensis* and *A. ipaensis*, indicating a gene loss in the two ancestral species of peanut. Compared with peanut, a homologous gene of *AdNRAMP2* in *A. duranensis* and a homologous gene of *AiNRAMP3* in *A. ipaensis* have got lost since the allotetraploidization event. Interestingly, two paralogous genes of peanut *AhNRAMP6* (*AhNRAMP6.3*/*6.4*) are located in different chromosomes (Chr.18 and Chr.19). However, their orthologous genes (*AiNRAMP6.1*/*6.2*) in *A. ipaensis* located in the same chromosome (B09). This might be a result of the translocation between chromosome B08 and B09 in *A. ipaensis* after the allotetraploidization event. Allotetraploid seems to have a higher capacity for avoiding gene loss than their diploid progenitors, and peanut is closer to *A. duranensis* than *A. ipaensis* in terms of the *NRAMP* family.

All *AhNRAMP* genes were collinear with the seven *PvNRAMPs* of the common bean through *AdNRAMP*s (Figure 3C), suggesting a close evolutionary relationship. Most of the *AdNRAMP*s were collinear with five *AtNRAMP* genes of *Arabidopsis* through *PvNRAMPs*, forming continuous colinear gene pairs. However, *AdNRAMP1.1*/*1.2* and *AdNRAMP2.1* did not form continuous colinear gene pairs with *AtNRAMP* genes (Figure 3C). These data suggest that, besides the common ancestor shared with *Arabidopsis*, some *NRAMP* genes of leguminous plants such as *AdNRAMP1.1*/*1.2* and *AdNRAMP2.1* in *A. duranensis* and *PvNRAMP7* and *PvNRAMP5* in common bean, might originate from other ancestors.

Gene duplication contributes to the evolution of novel genes with new functions [55]. However, newly duplicated genes are usually functionally redundant, which may lead to gene loss [56]. Thus, expression reduction facilitates the retention of duplicates and the conservation of their ancestral functions [56]. In the present study, seven *AhNRAMP* genes showed low expression in most of the 22 peanut tissues under normal conditions (Figure 6). Similar results have been reported in other gene families in peanut [49,50,51]. These results support the speculation that reduced expression might be beneficial to retain duplicate genes and their functional redundancy.

The NRAMPs typically consists of around 500 amino acid residues proteins with 10–12 TMDs in different species [10,42]. All PvNRAMPs from common beans have 12 TMDs, and the number of amino acid residues ranged from 507 to 554 [44]. The cacao NRAMP proteins are comprised of 510 to 557 amino acids with 11–13 TMDs [47]. In the current study, we found that, except that AhNRAMP1.2 contains 4 TMDs, all other AhNRAMP proteins contain 12 TMDs with amino acid numbers ranging from 498 to 560 (Table 1). According to syntenic analysis, *AhNRAMP1.2* might be derived from *AdNRAMP1.2* of the *A. duranensis* genome, which consists of 333 amino acids with 7 TMDs. The motif composition and exon/intron structure showed that AhNRAMP1.2 only contained the last two motifs (Motif 5 and 9), and the corresponding gene contained the last three exons. Multiple alignments revealed that AhNRAMP1.2 only contains the last five residues (MQGFL) of typical conserved amino acid residues GQSSTITGTYAGQY(/F)V(/I)MQ(/G/E)GFL. Thus, the gene was continuously shortened during evolution, and only the C-terminal fragment of *AhNRAMP*s was retained. However, the preferential expression of *AhNRAMP1.2* in stamens and the induction by Fe deficiency in roots confirm its roles in stamen development and Fe nutrition. Short NRAMP protein sequences have been reported in several plant species, such as *C. sinensis* and *Brassica napus* [46,57].

All AhNRAMP proteins were well modeled with the 3D model template, 5m94.1.A, which is the crystal structure of a divalent metal transporter from *Staphylococcus capitis* (ScaDMT). ScaDMT is a close prokaryotic homolog of the SLC11 (NRAMP) family that transports divalent transition-metal ions such as Fe^2+^, Mn^2+^, and Cd^2+^ across cellular membranes [58]. It has been confirmed that D49 and N52 from TM1, M226 from the side chain, and A223 from TM6 coordinate the Mn^2+^ substrate [58]. Moreover, the Mn^2+^ binding site also bound other cations such as Fe^2+^, Co^2+^, Ni^2+^, Cd^2+^, and Pb^2+^, while the position of Cu^2+^ shifted somewhat but retained a close interaction with M226 [58]. Surprisingly, Zn^2+^ is bound in another site where it closely interacts with H233 in TM6b [58]. Multiple sequence alignment showed considerable homology between the ScaDMT and AhNRAMP proteins throughout the TMDs (36.52–42.19% sequence identity). All AhNRAMPs contained the typical conserved amino acid residues GQSSTITGTYAGQY(/F)V(/I)MQ(/G/E)GFL except AhNRAMP1.2, in which the last five residues (MQGFL) were reserved. The conserved amino acid residues were located between the TM8 and TM9 [40]. Besides, conserved amino acid residues were also found in different TMDs, e.g., DPGN in TM1, EVIGS(/T)AI(/L)A in TM3, M(/T)PHNV(/L) FLHSS(/A)LVQ(/L)SF in TM6, A(/G)LLAS(/A) in TM8 (Figure 5). The similar structures indicate that AhNRAMPs might have equivalent physiological functions as ScaDMT.

Phylogenetic analysis grouped the NRAMP proteins into two subfamilies, and each of them was further divided into two groups (Figure 1). The classification concurred with those reported in previous studies [11,43,44,46]. Subfamily I was formed by eight AhNRAMP proteins (AhNRAMP2.1/2.2/2.3/2.4 and AhNRAMP3.1/3.2/3.3/3.4), and subfamily II contained seven other AhNRAMP members (AhNRAMP1.1/1.2/1.3 and AhNRAMP6.1/6.2/6.3/6.4). The two subfamilies differed from each other in pI, conserved motif composition, and subcellular localization of AhNRAMP proteins, as well as exons/intron structure. These traits, however, were highly conserved within a subfamily. Subfamily I AhNRAMPs are acid proteins (pI ranged from 4.92 to 5.42) with a specific conserved motif (Motif 7), while subfamily II members are basic proteins (pI ranged from 8.62 to 8.93) sharing another specific conserved motif (Motif 10) except AhNRAMP1.2 (Table 1, Figure 2A). Subfamily I genes contained four exons, while those in subfamily II had 13 exons except *AhNRAMP1.2* (Figure 2C). The subfamily I proteins were predicted to be localized to the vacuole membrane, while the subfamily II proteins were localized to the plasma membrane (Table 1). Similar results were reported by Qin et al. [43] in soybean. Indeed, experimental work confirmed that four subfamily I members (GmNRAMP1a, GmNRAMP2a, GmNRAMP2b, and GmNRAMP3a) from soybean were localized to the tonoplast, and two subfamily II members (GmNRAMP5a and GmNRAMP7) localized to the plasma membrane [43].

The *cis*-acting elements are important molecular switches that play a critical role in gene expression in plants. In the promoter regions of *AhNRAMP*s, many light-responsive elements (Box 4, AT1-motif, G-box, GT1-motif, and TCT-motif), stress-responsive elements (ARE, LTR, MBS, TC-rich repeats) and hormone-responsive elements (ABRE, CGTCAmotif, TGACG-motif, and TCA-element) were frequently identified, indicating that *AhNRAMP* genes might be regulated by complex regulatory networks.

Several homologous genes belonging to the subfamily I have been functionally characterized in *Arabidopsis*. AtNRAMP2 is localized to the *trans*-Golgi network and exports Mn from the *trans*-Golgi network lumen to the cytosol under Mn-deficient conditions [14,15]. AtNRAMP3 and AtNRAMP4 are vacuolar membrane-localized proteins that contribute to Fe and Mn nutrition by remobilizing vacuolar Fe stores [16,17,18]. Overexpression of *AtNRAMP3* leads to Cd hypersensitivity and increased accumulation of Fe and Cd in *Arabidopsis* [59]. In contrast, more subfamily II genes have been characterized in *Arabidopsis* and rice. OsNRAMP1/4/5/6 has been demonstrated to be plasma membrane-localized transporters in rice [20,21,22,23,24,26,27,28,32,33,34]. OsNRAMP1 and OsNRAMP5 are responsible for the uptake of Fe, Mn, and Cd in rice [20,21,22,23,24,26,27,28]. OsNRAMP6 function as Fe and Mn transporters and contribute to disease resistance in rice [34]. OsNRAMP4 transport Al but not for divalent cations in yeast [32,33]. AtNRAMP1 is a plasma membrane-localized Mn transporter essential for *Arabidopsis* growth in low-Mn conditions [12]. AtNRAMP6 is an intracellular metal transporter, and its overexpression increases Cd sensitivity in *Arabidopsis* [19]. OsNRAMP3 is a vascular bundles-specific Mn transporter that regulates the distribution of Mn in plants [30,31]. These studies fully demonstrated that the *NRAMP* gene family participates in the uptake and transport of divalent metals, including Fe, Mn, and Cd, in plants.

In peanut, most of the *AhNRAMP* genes are predominantly expressed in stamens and roots and/or developing seeds (Figure 6). Similar results have been reported in potato [41]. The results suggest that the *AhNRAMP* family genes might play an essential role in metal uptake by the roots and the transport of divalent metals to stamens and/or seeds during their developing processes. The expression of almost all *AhNRAMP* genes is induced by Fe deficiency in peanut roots (Figure 7). The up-regulation of *AhNRAMP* genes implies a high capacity for Fe uptake. However, significant but negative correlations were found between the expression of *AhNRAMP* genes in roots and Fe accumulation in peanut plants (Table 3). Such a contradictory result is due to the Fe-deficient plants having no absorbable Fe despite their strong ability to take up Fe.

In agreement with previous studies [6,7,38], our results indicate that Fe deficiency significantly increases the uptake and accumulation of Cd in peanut plants (Figure 8). Moreover, the uptake and accumulation of Cd in peanut plants are positively correlated with the expression of all *AhNRAMP* genes except *AhNRAMP2.4* and/or *AhNRAMP3.2* (Table 3). These results indicated that the induction of *AhNRAMP*s by Fe deficiency in peanut roots might be, at least partially, responsible for the increased Cd accumulation in Fe-deficient plants.

Not unexpectedly, we found that Silihong showed a higher capacity for uptake and translocation of Cd from roots to shoots than Fenghua 1 under Fe-deficient conditions (Figure 8). This result is consistent with our previous studies [37,49,51]. Similarly, the expression of all *AhNRAMP* genes except *AhNRAMP2.4* was higher in the roots of Silihong than that of Fenghua 1 in the treatments of Fe deficiency with Cd exposure (Figure 7). It seems that the higher expression of *AhNRAMP* genes contributes to the high capacity for Cd accumulation in Silihong compared with Fenghua 1.

## 4. Materials and Methods

### 4.1. Plant Materials and Treatments

Two peanut cultivars, Fenghua 1 (Fe deficiency sensitive/Cd tolerant cultivar) and Silihong (Fe deficiency tolerant/Cd sensitive cultivar) were used for experiments [37]. Seeds were surface sterilized with 5% sodium hypochlorite, rinsed in deionized water for 24 h in the dark, and then sown in the sand for germination. After germination, 3-d-old uniform seedlings were transplanted to polyethylene pots and cultured in hydroponics under controlled conditions for a week. The culture conditions and nutrient solutions were followed as described previously by Su et al. [38]. Ten-day-old seedlings were exposed to 0 or 2 μM CdCl_2_ under Fe-sufficient (50 μM Fe-EDTA) or Fe-deficient (0 μM Fe-EDTA) conditions, respectively. Each treatment had three biological replicates with three seedlings per replication. During the growing period, nutrient solutions were renewed twice a week. Plants were harvested following 14 days of treatment. Fresh root tissues were immediately frozen in liquid nitrogen and stored at −80 °C for RT-qPCR analysis.

### 4.2. Determination of Fe and Cd in Peanut Plants

The harvested plants were separated into roots and shoots. The root was rinsed with 20 mM Na_2_EDTA for 15 min to remove the surface-bound metal ions. Thereafter, roots and shoots were oven-dried, weighed, and ground into powder. After digestion with HNO_3_–HClO_4_ (3:1, *v*/*v*), Cd and Fe concentrations in the samples were determined by flame atomic absorbance spectrometry (WFX-110, Beijing Rayleigh Analytical Instrument Company, Beijing, China). The total Fe/Cd in plants and the percentage of Fe/Cd in shoots were calculated using the equations reported by Liu et al. [37].

### 4.3. Identification and Bioinformatics Analyses of NRAMP Family in Peanut

The sequences of six AtNRAMP (AtNRAMP1-6) and seven OsNRAMP proteins (OsNRAMP1-7) were used as queries for TBLASTP against the genome of peanut (cv. Tifrunner) using TBtools software v. 1. 108 [60]. After examining for the presence of the conserved NRAMP domain (PF01566) using the hmmscan tool (https://www.ebi.ac.uk/Tools/hmmer/search/hmmscan, accessed on 6 July 2022), candidates were further aligned by ClustalW, along with AtNRAMPs and OsNRAMPs. An NJ (neighbor-joining) phylogenetic tree was constructed with the Poisson model and 1000 bootstraps using the MEGA-X program (v. 10.2.6). Sequences clustered together with AtNRAMPs and OsNRAMPs could be recognized as AhNRAMPs. To investigate the phylogenetic relationships of AhNRAMPs, full-length sequences of NRAMP proteins from peanut, *Arabidopsis*, rice, soybean, and common beans were used to construct a phylogenetic tree as the method mentioned above. The phylogenetic tree was displayed and modified using iTOL (https://itol.embl.de/itol.cgi, accessed on 20 July 2022).

The physical and chemical properties of AhNRAMP proteins were analyzed using the ProtParam tool (https://web.expasy.org/protparam/, accessed on 6 July 2022) [61]. The number of TMDs in proteins was predicted using TOPCONS (http://topcons.net/, accessed on 8 June 2022) [62]. Subcellular localization of AhNRAMP proteins was predicted using ProtComp 9.0 (http://linux1.softberry.com/berry.phtml?topic=protcomppl&group=programs&subgroup=proloc, accessed on 30 June 2022). The conserved domains and motifs were analyzed using the Pfam tool (http://pfam.xfam.org/search#tabview=tab1, accessed on 15 July 2022) and the MEME v. 5.3.3 (https://meme-suite.org/meme/tools/meme, accessed on 20 July 2022), respectively [63,64]. Homology-modeled 3D structures of AhNRAMP proteins were predicted using the SwissModel (https://swissmodel.expasy.org/, accessed on 20 September 2022) [65].

The exon/intron structures were detected with genomic and coding sequences using GSDS v. 2.0 (http://gsds.gao-lab.org/, accessed on 10 June 2022) [66]. The synteny relationship of *NRAMP* genes within peanut genome and among genomes of multiple species, including peanut, *A. duranensis*, *A. ipaënsis*, *Arabidopsis*, and common bean, was analyzed using one-step MCScanX integrated into the TBtools software v. 1. 108 [60]. Gene collinearity and Ka/Ks (ratio of nonsynonymous substitution rate to synonymous substitution rate) were analyzed by one-step MCScanX and simple *Ka*/*Ks* calculator (NJ) in TBtools software v. 1. 108, respectively [60]. The *cis*-acting elements in the promoter sequences (upstream 2.0 kb) were predicted using PlantCARE (http://bioinformatics.psb.ugent.be/webtools/plantcare/html/, accessed on 1 January 2023) [67].

### 4.4. Gene Expression Analysis Based on RNA-Seq Data

Expression profiles of *AhNRAMP* genes in different tissues were identified using RNA-seq data of cv. Tifrunner obtained from PeanutBase (https://www.peanutbase.org/, accessed on 15 June 2022) [68]. The expression of *AhNRAMP* genes was normalized and represented in TPM (transcripts per kilobase of exon model per million mapped reads), and lg(TPM + 1) was used to construct the heatmap diagram.

### 4.5. RNA Extraction and qRT-PCR Analysis

Total RNA was extracted from the roots using a plant RNA rapid extract Kit (Coolaber, Beijing, China). First-strand cDNA was synthesized using MonScript™ RTIII Super Mix with dsDNase (Monad, Suzhou, China) according to the manufacturer’s protocol. For qRT-PCR analysis, specific primers of *AhNRAMP* genes were designed, and *60S* ribosomal protein L7-2 (NCBI_ID: 112697914) was used as an internal control (Appendix A). qRT-PCR was performed using 10× diluted cDNAs and MonAmp™ ChemoHS Plus qPCR Mix (Monad, Suzhou, China) on the LightCycler^®^96 Instrument (Roche Applied Science, Indianapolis, IN, USA). Each sample was repeated with three technical replicates. The relative gene expression was calculated using the delta-delta CT method (2^−ΔΔCT^) [69].

### 4.6. Statistical Analysis

Data were subjected to a one-way analysis of variance, and significant variations among means were determined by Duncan’s Multiple Range Test (*p* < 0.05). Pearson’s correlation analysis was performed to determine the correlations between gene expression and Fe/Cd accumulation. All statistical analysis was conducted using IBM SPSS Statistics v.22 (IBM, New York, NY, USA).

## 5. Conclusions

A total of 15 *AhNRAMP* genes were identified from the peanut genome, including seven WGD-derived gene pairs and a segmental duplicated gene. Phylogenetic analysis grouped the NRAMP proteins into two distinct subfamilies. Subfamily I contains eight acid proteins with the specific motif 7, while subfamily II includes seven basic proteins sharing the specific motif 10. Subfamily I genes contained four exons, while those in subfamily II had 13 exons. The subfamily I proteins were predicted to localize in the vacuole membrane, while the subfamily II proteins localized to the plasma membrane. All AhNRAMP proteins are perfectly modeled on the 5m94.1.A template, suggesting that AhNRAMPs might have equivalent physiological functions as ScaDMT. Most *AhNRAMP* genes are preferentially expressed in roots, stamens, and developing seeds, suggesting an essential role in metal uptake and their transport to stamens and developing seeds. The expression of most *AhNRAMP*s is induced by Fe deficiency in peanut roots and positively correlated with Cd accumulation, indicating a crucial role in Fe/Cd interactions in peanuts. The findings provide essential information to understand the functions of the *AhNRAMP* genes in the Fe/Cd interactions in peanut plants, which are of great importance for screening or breeding cultivars for improving Fe nutrition while reducing Cd accumulation.

## Figures and Tables

**Figure 1 ijms-24-01713-f001:**
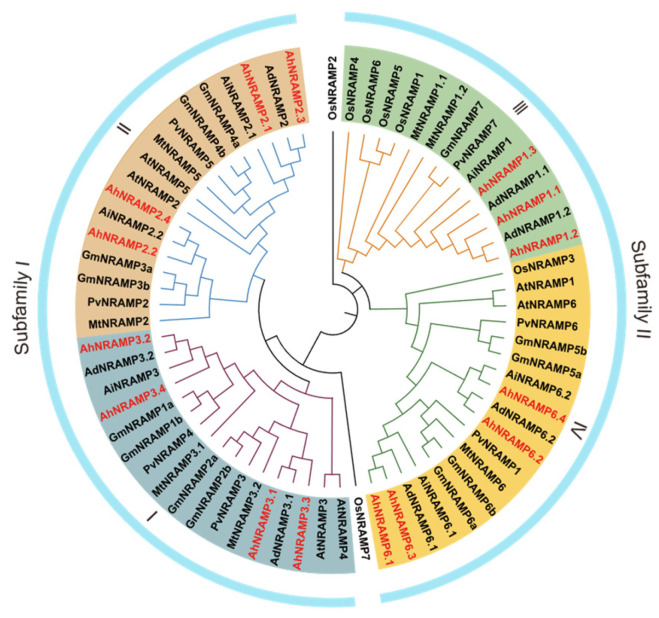
Phylogenetic relationships of NRAMP proteins from peanut and other plant species. The 15 AhNRAMP proteins of peanut were marked in red color. The species involved in the evolutionary tree include *Arabidopsis thaliana* (AtNRAMPs), *Oryza sativa* (OsNRAMPs), *Glycine max* (GmNRAMPs), *A. duranensis* (AdNRAMPs), *A. ipaensis* (AiNRAMPs), *Medicago truncatula* (MtNRAMPs), and *Phaseolus vulgaris* (PvNRAMPs).

**Figure 2 ijms-24-01713-f002:**
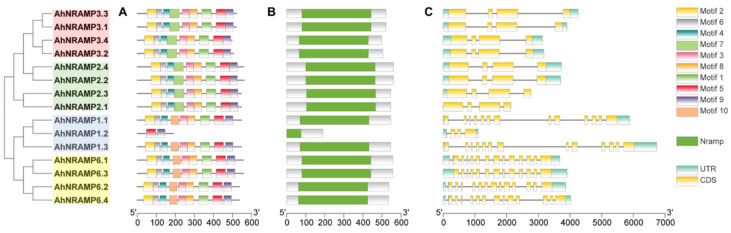
Conserved motifs (**A**) and domains (**B**) in AhNRAMP proteins as well as exon-intron structure (**C**) of *AhNRAMP* genes from peanut. UTR and CDS represent untranslated regions and coding sequences, respectively.

**Figure 3 ijms-24-01713-f003:**
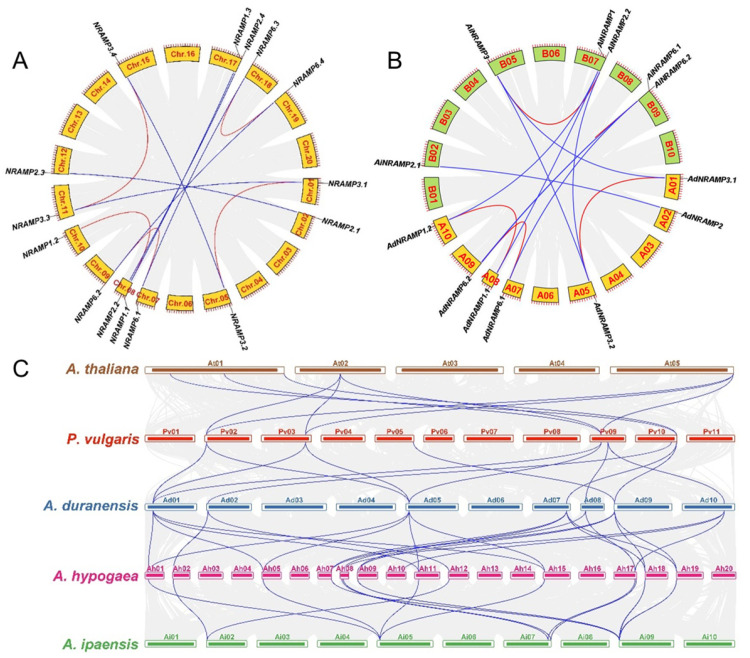
Synteny relationship of *NRAMP* gene pairs in peanut and other four species. (**A**) Synteny relationship of *AhNRAMP* gene pairs in peanut. (**B**) Synteny relationship of *NRAMP* gene pairs between *A. duranensis* and *A. ipaensis*. (**C**) Synteny relationship of *NRAMP* gene pairs among five plant species. The red and blue lines represent segmental duplicated genes and synteny genes, respectively. The gray lines show the collinear blocks of the plant genomes.

**Figure 4 ijms-24-01713-f004:**
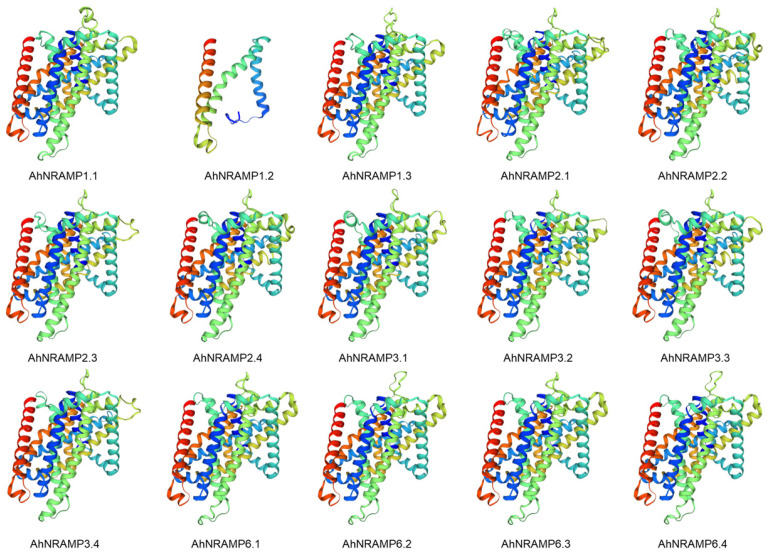
Predicted 3D structure of peanut AhNRAMP proteins by SwissModel. Models were visualized by rainbow color from N to C terminus.

**Figure 5 ijms-24-01713-f005:**
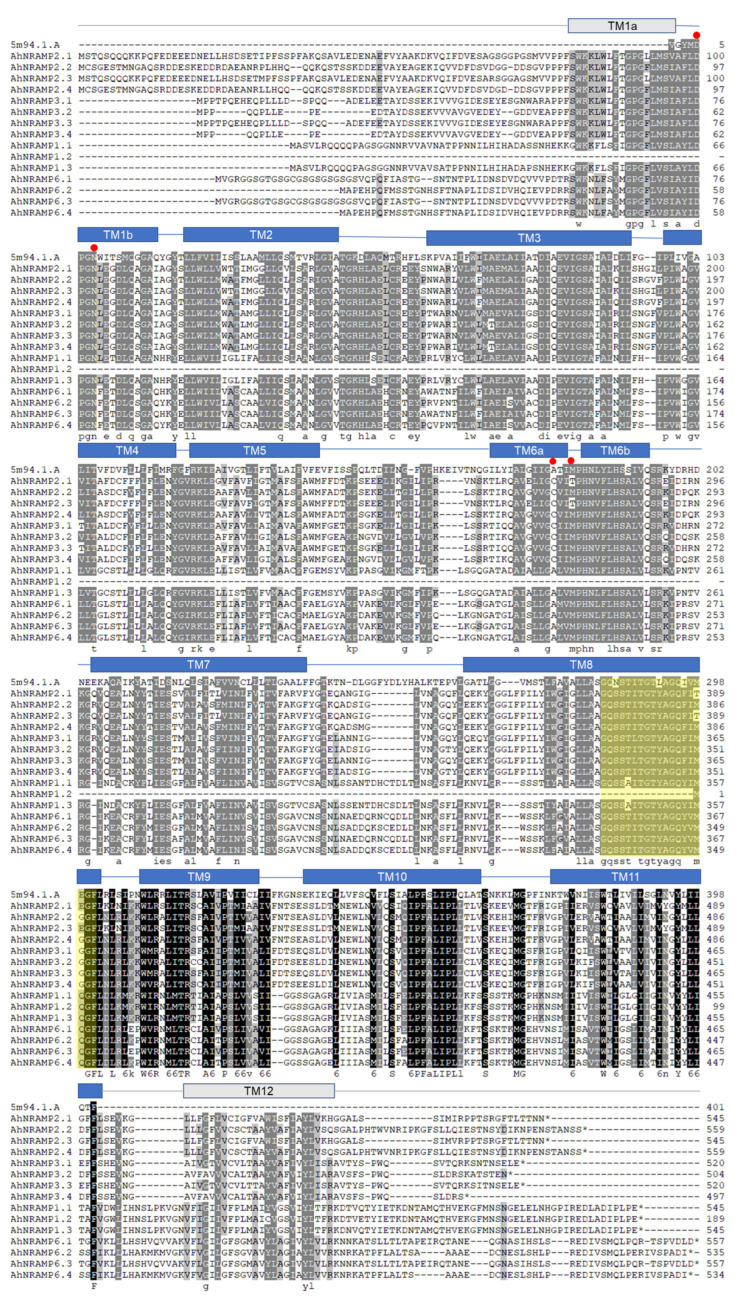
Sequence alignment of ScaDMT and AhNRAMP with ClustalW. Identical residues are highlighted in black and homologous residues in gray, while the typical conserved amino acid residues GQSSTITGTYAGQY(/F)V(/I)MQ(/G/E)GFL are highlighted in yellow. Secondary structure elements are shown above the sequences. The red circles indicate residues of the transition metal binding site in ScaDMT.

**Figure 6 ijms-24-01713-f006:**
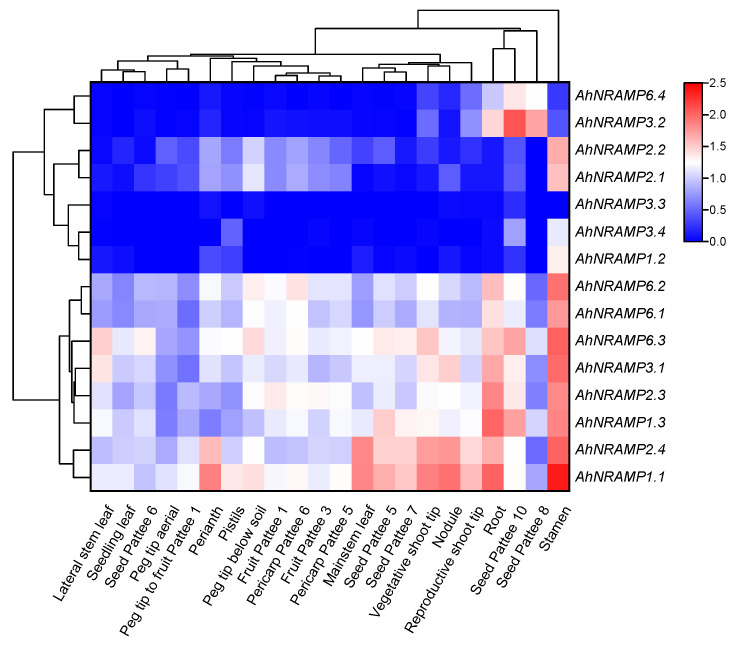
Tissue-specific expression profiles of *AhNRAMP* genes in peanut. Gene expression is expressed in lg(TPM + 1). Pattee 1, 3, 5, 6, 7, 8, and 10 represent different developmental stages of peanut pods based on Pattee’s classification.

**Figure 7 ijms-24-01713-f007:**
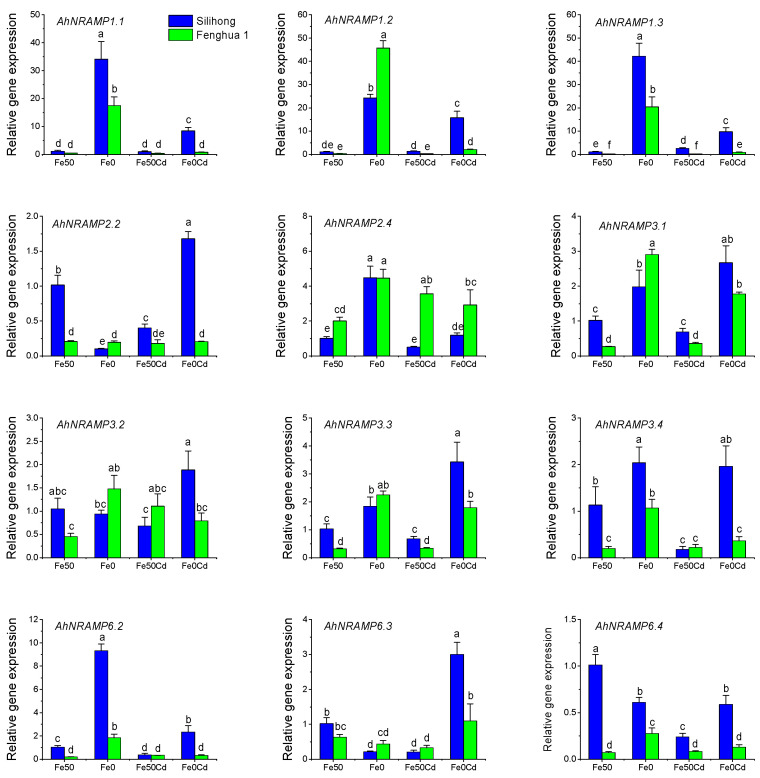
Expression levels of 12 *AhNRAMP* genes in peanut roots exposed to 0 or 2 μM Cd under Fe-sufficient (Fe50) and Fe-deficient (Fe0) conditions for 14 days. Data (means ± SE, n = 3) sharing the same letter(s) above the error bars are not significantly different at the 0.05 level based on the Duncan multiple range test.

**Figure 8 ijms-24-01713-f008:**
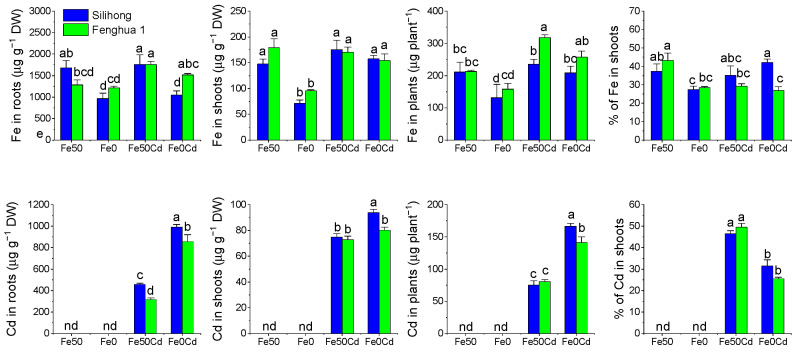
Fe/Cd accumulation and translocation in two peanut cultivars exposed to 0 or 2 μM Cd under Fe-sufficient (Fe50) and Fe-deficient (Fe0) conditions for 14 days. Data (means ± SE, n = 3) sharing the same letter(s) above the error bars are not significantly different at the 0.05 level based on the Duncan multiple range test. nd, not determinable.

**Table 1 ijms-24-01713-t001:** Molecular characterization of *AhNRAMP* genes and corresponding proteins in peanut.

Gene Name	Gene ID	Gene Length (bp)	CDS (bp)	Aa ^a^	MW ^b^ (kDa)	Instability	Aliphatic Index	GRAVY ^c^	pI ^d^	No. of TMD ^e^	Location
*AhNRAMP1.1*	112706038	2062	1638	546	58.78	36.06	120.26	0.564	8.93	12	PM ^f^
*AhNRAMP1.2*	112717764	742	570	190	20.83	31.17	123.76	0.584	8.74	4	PM
*AhNRAMP1.3*	112766867	2393	1638	546	58.77	37.32	120.79	0.566	8.93	12	PM
*AhNRAMP2.1*	112717946	1638	1638	546	60.18	41.85	111.45	0.494	5.21	12	VM ^g^
*AhNRAMP2.2*	112706457	2334	1680	560	61.35	35.46	108.53	0.355	5.35	12	VM
*AhNRAMP2.3*	112729510	1756	1638	546	60.24	42.40	110.55	0.486	5.29	12	VM
*AhNRAMP2.4*	112764447	2341	1680	560	61.39	34.44	108.87	0.363	5.27	12	VM
*AhNRAMP3.1*	112799830	1868	1563	521	57.34	38.79	119.81	0.530	5.23	12	VM
*AhNRAMP3.2*	112800530	2085	1515	505	55.15	39.60	122.66	0.667	4.94	12	VM
*AhNRAMP3.3*	112720549	1984	1563	521	57.37	37.77	119.25	0.532	5.42	12	VM
*AhNRAMP3.4*	112748442	2056	1494	498	54.41	41.17	124.19	0.693	4.92	12	VM
*AhNRAMP6.1*	112703806	2277	1674	558	59.78	31.48	118.01	0.580	8.62	12	PM
*AhNRAMP6.2*	112709742	2141	1608	536	58.14	38.23	118.52	0.598	8.69	12	PM
*AhNRAMP6.3*	112773194	2505	1674	558	59.78	31.48	118.01	0.580	8.62	12	PM
*AhNRAMP6.4*	112776188	1869	1605	535	58.04	38.65	118.56	0.598	8.57	12	PM

^a^ amino acid number, ^b^ molecular weight, ^c^ grand average of hydropathicity, ^d^ isoelectric points, ^e^ transmembrane domain, ^f^ plasma membrane, ^g^ vacuole membrane.

**Table 2 ijms-24-01713-t002:** *Ka*/*Ks* analysis of all gene duplication pairs for *AhNRAMP* genes.

Gene Pairs	Duplicate Type	*Ka* ^a^	*Ks* ^b^	*Ka*/*Ks* ^c^	Positive Selection	Divergence Time (Mya)
*AhNRAMP1.1*/*1.2*	Segmental	0.0069	0.0077	0.9017	No	0.472
*AhNRAMP3.1*/*3.2*	Segmental	0.0966	0.9455	0.1022	No	58.221
*AhNRAMP3.3*/*3.4*	Segmental	0.0865	0.8961	0.0966	No	55.178
*AhNRAMP6.1*/*6.2*	Segmental	0.1076	0.8321	0.1294	No	51.236
*AhNRAMP6.3*/*6.4*	Segmental	0.1074	0.8417	0.1276	No	51.826
*AhNRAMP1.1*/*1.3*	Whole-genome	0.0065	0.0201	0.3237	No	1.240
*AhNRAMP2.1*/*2.3*	Whole-genome	0.0040	0.0481	0.0834	No	2.963
*AhNRAMP2.2*/*2.4*	Whole-genome	0.0047	0.0329	0.1440	No	2.023
*AhNRAMP3.1*/*3.3*	Whole-genome	0.0051	0.0368	0.1399	No	2.265
*AhNRAMP3.2*/*3.4*	Whole-genome	0.0009	0.0219	0.0407	No	1.351
*AhNRAMP6.1*/*6.3*	Whole-genome	0.0000	0.0000	-	No	0.000
*AhNRAMP6.2*/*6.4*	Whole-genome	0.0008	0.0600	0.0136	No	3.697

^a^ The number of nonsynonymous substitutions per nonsynonymous site, ^b^ the number of synonymous substitutions per synonymous site, ^c^ *Ka*/*Ks* ratios.

**Table 3 ijms-24-01713-t003:** Correlations Fe/Cd accumulation and the expression of *AhNRAMP* genes in the roots of two peanut cultivars (*n* = 24 and 12 for Fe and Cd accumulation, respectively).

Gene Expression	Fe in Roots	Fe in Shoots	Total Fe in Plants	% of Fe in Shoots	Cd in Roots	Cd in Shoots	Total Cd in Plants	% of Cd in Shoots
*AHNRAMP1.1*	−0.646 **	−0.820 **	−0.681 **	−0.294	0.699 *	0.785 **	0.700 *	−0.432
*AHNRAMP1.2*	−0.541 **	−0.719 **	−0.588 **	−0.278	0.699 *	0.876 **	0.738 **	−0.380
*AHNRAMP1.3*	−0.610 **	−0.828 **	−0.673 **	−0.352	0.650 *	0.754 **	0.616 *	−0.367
*AHNRAMP2.2*	−0.142	0.223	−0.015	0.529 **	0.657 *	0.830 **	0.656 *	−0.326
*AHNRAMP2.4*	−0.349	−0.544 **	−0.230	−0.498 *	−0.183	−0.261	−0.001	−0.026
*AHNRAMP3.1*	−0.570 **	−0.555 **	−0.501 *	−0.123	0.888 **	0.802 **	0.871 **	−0.721 **
*AHNRAMP3.2*	−0.192	−0.192	−0.099	0.069	0.423	0.718 **	0.505	−0.093
*AHNRAMP3.3*	−0.594 **	−0.400	−0.438 *	0.010	0.847 **	0.752 **	0.817 **	−0.705 *
*AHNRAMP3.4*	−0.561 **	−0.568 **	−0.546 **	0.051	0.670 *	0.809 **	0.678 *	−0.364
*AHNRAMP6.2*	−0.596 **	−0.740 **	−0.610 **	−0.235	0.603 *	0.748 **	0.622 *	−0.285
*AHNRAMP6.3*	−0.261	0.246	0.055	0.464 *	0.808 **	0.908 **	0.872 **	−0.550
*AHNRAMP6.4*	−0.178	−0.331	−0.445 *	0.158	0.640 *	0.761 **	0.596 *	−0.367

* *p* < 0.05, ** *p* < 0.01.

## Data Availability

The datasets used and/or analyzed during the current study are available from the corresponding author upon reasonable request.

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
