# Peer review of "Genome-Wide Identification and Expression Analysis Reveals Roles of the NRAMP Gene Family in Iron/Cadmium Interactions in Peanut"

_ijms, 2023, doi:10.3390/ijms24021713_

Round 1

Reviewer 1 Report

In retrospect, I do not feel I am qualified to review this paper as it lies outside my area of expertise but I will defer to the editor for a final judgement on that matter. In particular, much of the paper is simply descriptors of this gene family in terminology I am unfamiliar with and not used in my line of work.

The novelty of the research is average. I have seen many similar papers that look at different gene families for different traits in different species. I am generally unsure of what the applicability of these types of studies are. The research approach is: “If this gene family played this role in one species, then it might play a role in this other species.” I am not sure if this is the soundest approach due to its implicit bias. Furthermore, just because the genes of interest are differentially expressed in response to the stress, does not necessarily mean the causal polymorphism lies in or anywhere near the genes of interest. The causal polymorphism could lie in an upstream regulator of these genes. In either case, molecular markers would need to be developed off or near this causal polymorphism in order to breed for the trait of interest. No attempt was made to look for such polymorphisms between the two cultivars studied.

No data (even a visual rating) was taken to describe the injury severity of any plant under any treatment. While the amount of Fe and Cd was quantified, there is no independent proof that the Fe levels were low enough and/or the Cd levels high enough to have a significant adverse effect on plant growth. Furthermore, plants were harvested after 24 days, well short of the time it takes a peanut plant to mature. I understand that this was done hydroponically and peanut has a difficult growth habit to study but there is no effort to address whether the induced stresses would impact yield. There is no data presented that the applied treatments had obvious effects on plant growth.

It seems like it would have made more sense to use the hydroponic system to screen a wider variety of germplasm to get a better understanding of the breadth of phenotypic diversity.

If the two selected cultivars represent the available extremes for Fe/Cd response, why not cross them, develop a segregating population and map the causal loci?

Both the abstract (line 20) and conclusion (line 530) mention the ‘5m94.1.A template’. Despite a brief description in the discussion (line 364), I am unsure what this is or why this is such a major finding.

The AhNRAMP genes seem to be disproportionately expressed in stamens but no explanation is offered as to why.

Author Response

Response to Review1

In retrospect, I do not feel I am qualified to review this paper as it lies outside my area of expertise but I will defer to the editor for a final judgement on that matter. In particular, much of the paper is simply descriptors of this gene family in terminology I am unfamiliar with and not used in my line of work.

Response: Thanks for your comments.

The novelty of the research is average. I have seen many similar papers that look at different gene families for different traits in different species. I am generally unsure of what the applicability of these types of studies are. The research approach is: “If this gene family played this role in one species, then it might play a role in this other species.” I am not sure if this is the soundest approach due to its implicit bias. Furthermore, just because the genes of interest are differentially expressed in response to the stress, does not necessarily mean the causal polymorphism lies in or anywhere near the genes of interest. The causal polymorphism could lie in an upstream regulator of these genes. In either case, molecular markers would need to be developed off or near this causal polymorphism in order to breed for the trait of interest. No attempt was made to look for such polymorphisms between the two cultivars studied.

Response: Thanks for your comments. I totally agree with you. However, we must admit that our existing knowledge is still unable to understand these diverse regulatory mechanisms. Before revealing these regulatory mechanisms, we should firstly identify key genes involved in plant biological processes. Our findings provide essential information to understand the functions of AhNRAMPs in the iron/cadmium interactions in peanuts.

No data (even a visual rating) was taken to describe the injury severity of any plant under any treatment. While the amount of Fe and Cd was quantified, there is no independent proof that the Fe levels were low enough and/or the Cd levels high enough to have a significant adverse effect on plant growth. Furthermore, plants were harvested after 24 days, well short of the time it takes a peanut plant to mature. I understand that this was done hydroponically and peanut has a difficult growth habit to study but there is no effort to address whether the induced stresses would impact yield. There is no data presented that the applied treatments had obvious effects on plant growth.

Response: Thanks for your comments. According to your suggestion, we have included biomass and leaf chlorophyll content (SPAD value) data, which were shown in Figure S3. Some statements were also included in the text (p.8, lines 305-312). The purpose of the experiment is to determine the relationship between Cd/Fe accumulation and gene expression. You should know that gene expression was changed by stresses quickly, and it is varied among different developmental stages. Thus, long-term experiment is not suitable for this study. Indeed, we have carried some long-term experiments several years ago. Several related papers are as following:

Su G, Li F, Lin J, Liu C, Shi G. Peanut as a potential crop for bioenergy production via Cd-phytoextraction: A life-cycle pot experiment. Plant and Soil, 2013, 365(1-2):337-345.   

Xia S, Wang X, Su G, Shi G. Effects of drought on cadmium accumulation in peanuts grown in a contaminated calcareous soil. Environmental Science and Pollution Research, 2015, 22(23): 18707-18717.

Shi G, Su G, Lu Z, Liu C, Wang X. Relationship between biomass, seed components and seed Cd concentration in various peanut (Arachis hypogaea L.) cultivars grown on Cd-contaminated soils. Ecotoxicology and Environmental Safety, 2014, 110: 174-181.

It seems like it would have made more sense to use the hydroponic system to screen a wider variety of germplasm to get a better understanding of the breadth of phenotypic diversity.

Response: Thanks for your comments. The purpose of the experiment is to determine the relationship between Cd/Fe accumulation and gene expression. We have done such research several years ago. Several related papers are as following:

Lu Z, Zhang Z, Su Y, Liu C, Shi G. Cultivar variation in morphological response of peanut roots to cadmium stress and its relation to cadmium accumulation. Ecotoxicology and Environmental Safety, 2013, 91: 147-155.

Su Y, Liu J, Lu Z, Wang X, Zhang Z, Shi G. Effects of iron deficiency on subcellular distribution and chemical forms of cadmium in peanut roots in relation to its translocation. Environmental and Experimental Botany, 2014, 97: 40-48.

Su Y, Wang X, Liu C, Shi G. Variation in cadmium accumulation and translocation among peanut cultivars as affected by iron deficiency. Plant and Soil, 2013, 363(1-2): 201-213.

Su Y, Zhang Z, Su G, Liu J, Liu C, Shi G. Genotypic differences in spectral and photosynthetic response of peanut to iron deficiency. Journal of Plant Nutrition, 2015, 38(1): 145-160.

If the two selected cultivars represent the available extremes for Fe/Cd response, why not cross them, develop a segregating population and map the causal loci?

Response: Thanks for your comments. Your suggestion is very constructive. We will do this work in the future.

Both the abstract (line 20) and conclusion (line 530) mention the ‘5m94.1.A template’. Despite a brief description in the discussion (line 364), I am unsure what this is or why this is such a major finding.

Response: Thanks for your comments. 5m94.1.A is a protein name in structural biology. Because no research was carried out on our AhNRAMP proteins, we matched these unknown protein sequences with proteins (templates) whose structure has been known. Thus, we can infer the possible functions according to templates.

The AhNRAMP genes seem to be disproportionately expressed in stamens but no explanation is offered as to why.

Response: Thanks for your comments. This has been explained in the manuscript (lines, 624-628; 648-650). 

Reviewer 2 Report

In this manuscript, the authors identified 15 natural resistance-associated macrophage protein (NRAMP) genes in Peanut using bioinformatics-based methods and also performed other analyses i.e., phylogenetics, gene duplication, Protein structure analysis, expression analysis etc. However, I have some crucial criticisms/comments (detailed below).

1. The introduction section should be improved. To make the introduction more informative, the authors should also discuss the detailed functional roles of some specific NRAMP genes in plants. 

2. In the phylogenetic analysis of NRAMP Genes, authors performed analysis using protein sequences from Arabidopsis, rice, Glycine max, and Phaseolus vulgaris only. They can draw a better picture/result if they use more species including some form Fabaceae family for phylogenetic analysis

3. In my opinion some more experiments should be conducted to further functional validations of results. The following experiments should be conducted, (a) Localization of some selected genes using GFP (b) functional validation of genes using transgenic approach/VIGS/CRISPR for drought stress.

4. The prediction of cis-acting regulatory elements should be performed. 

5. Include a figure that shows the localization of NRAMP genes on the chromosomes.

Author Response

Response to Review 2

In this manuscript, the authors identified 15 natural resistance-associated macrophage protein (NRAMP) genes in Peanut using bioinformatics-based methods and also performed other analyses i.e., phylogenetics, gene duplication, Protein structure analysis, expression analysis etc. However, I have some crucial criticisms/comments (detailed below).

Response: Thanks for your comments.

  1. The introduction section should be improved. To make the introduction more informative, the authors should also discuss the detailed functional roles of some specific NRAMP genes in plants.

Response: Thanks for your comments. According to your constructive suggestion, we have rewritten the introduction to discuss the detailed functional roles of some specific NRAMP genes, and some new literature was cited in the revised manuscript (lines 56-83).

  1. In the phylogenetic analysis of NRAMP Genes, authors performed analysis using protein sequences from Arabidopsis, rice, Glycine max, and Phaseolus vulgaris only. They can draw a better picture/result if they use more species including some form Fabaceae family for phylogenetic analysis

Response: Thank you for your constructive suggestion. We have re-plotted the phylogenetic tree, in which three Fabaceae family species such as A. duranensis, A. ipaensis, Medicago truncatula were included (Figure 1, lines 134-135; 146).

  1. In my opinion some more experiments should be conducted to further functional validations of results. The following experiments should be conducted, (a) Localization of some selected genes using GFP (b) functional validation of genes using transgenic approach/VIGS/CRISPR for drought stress.

Response: Thanks for your comments. Your suggestion is very constructive. We are currently conducting these studies.

  1. The prediction of cis-acting regulatory elements should be performed.

Response: Thanks for your comments. According to your constructive suggestion, we have performed the prediction of cis-acting regulatory elements, which were presented in Table S4. We tried to visualize the results, but did not obtain a good effect due to the length of elements (4-9 bp) is too short relative to promoter sequence (2000 bp). Considering that the table is too large, we put it in the supplementary materials. The results have been described and their implication has been discussed in the revised manuscript (lines 279-288; 500-505).

  1. Include a figure that shows the localization of NRAMP genes on the chromosomes.

Response: Thanks for your comments. According to your constructive suggestion, we have included a figure that shows the localization of NRAMP genes on the chromosomes. Because it is too large and is the same as Figure 3A to some extent, we put it in the supplementary materials (Figure S2).

Reviewer 3 Report

The manuscript is very well written. The authors have substantiated their observations appropriately. The manuscript, however, needs general proofreading for random grammatical and spelling mistakes. 

I wish my best to the authors.

Author Response

Response to Review 3

The manuscript is very well written. The authors have substantiated their observations appropriately. The manuscript, however, needs general proofreading for random grammatical and spelling mistakes.

Response: Thanks for your comments. According to your constructive suggestion, we have carefully checked and corrected the grammatical and spelling mistakes across the manuscript. All changes made to the text are marked up using the “Track Changes” function in the Word file of the revised manuscript.

Reviewer 4 Report

The manuscript meets the requirements for publication on the journal.

Particularly,

- The approach, wisely combining gene expression and genomics-level evidence, is coherent with the aim, and carefully detailed in 'Materials and Methods' section.

- The 'Discussion' section is quite detailed and sheds light on promising new hypotheses

As a whole, the work can be accepted for publication on the journal.

Author Response

Response to Review 4

The manuscript meets the requirements for publication on the journal. Particularly,

- The approach, wisely combining gene expression and genomics-level evidence, is coherent with the aim, and carefully detailed in 'Materials and Methods' section.

- The 'Discussion' section is quite detailed and sheds light on promising new hypotheses

As a whole, the work can be accepted for publication on the journal.

Response: Thank you very much for your comments.

Round 2

Reviewer 2 Report

The authors have clarified all the questions that I raised. Now I feel the manuscript could be accepted for publication.